# Bulbous Plants *Drimia*: “A Thin Line between Poisonous and Healing Compounds” with Biological Activities

**DOI:** 10.3390/pharmaceutics13091385

**Published:** 2021-09-01

**Authors:** Madira Coutlyne Manganyi, Gothusaone Simon Tlatsana, Given Thato Mokoroane, Keamogetswe Prudence Senna, John Frederick Mohaswa, Kabo Ntsayagae, Justine Fri, Collins Njie Ateba

**Affiliations:** 1Department of Biological and Environmental Sciences, Faculty of Natural Sciences, Walter Sisulu University, PBX1, Mthatha 5117, South Africa; 2Department of Microbiology, Mafikeng Campus, North West University, Mmabatho 2735, South Africa; tlatsanakgosi@gmail.com (G.S.T.); thatogven@gmail.com (G.T.M.); tshego.senna@gmail.com (K.P.S.); johnf6813@gmail.com (J.F.M.); kabokayrock1@gmail.com (K.N.); frijustine2000@gmail.com (J.F.); 3Food Security and Safety Niche Area, Faculty of Natural and Agricultural Sciences, North-West University, Mmabatho 2735, South Africa; Collins.Ateba@nwu.ac.za

**Keywords:** *Drimia*, *Urginea*, poisonous, compounds, toxicity, bulbous plants

## Abstract

*Drimia* (synonym *Urginea*) plants are bulbous plants belonging to the family Asparagaceae (formerly the family Hyacinthaceae) and are distinctive, powerful medicinal plants. Just some species are indigenous to South Africa and have been traditionally utilized for centuries to cure various diseases and/or ailments. They have been recognized among the most famous and used medicinal plants in South Africa. Traditionally, the plants are used for various illnesses such as dropsy, respiratory disease, bone and joint complications, skin disorders, epilepsy and cancer. A number of studies have reported biological properties such as antiviral, antibacterial, antioxidant and anti-inflammatory, immunomodulatory, and anticancer activities. Their bulbs are a popular treatment for colds, measles, pneumonia, coughs, fever and headaches. However, some plant species are regarded as one of the six most common poisonous plants in Southern Africa that are toxic to livestock and humans. Due to the therapeutic effects of the *Drimia* plant bulb, research has focused on the phytochemicals of *Drimia* species. The principal constituents isolated from this genus are cardiac glycosides. In addition, phenolic compounds, phytosterols and other phytochemical constituents were identified. This study constitutes a critical review of *Drimia* species’ bioactive compounds, toxicology, biological properties and phytochemistry, advocating it as an important source for effective therapeutic medicine. For this purpose, various scientific electronic databases such as ScienceDirect, Scopus, Google Scholar, PubMed and Web of Science were researched and reviewed to conduct this study. Despite well-studied biological investigations, there is limited research on the toxic properties and the toxic compounds of certain *Drimia* species. Searching from 2017 to 2021, Google Scholar search tools retrieved 462 publications; however, only 3 investigated the toxicity and safety aspects of *Drimia.* The aim was to identify the current scientific research gap on *Drimia* species, hence highlighting a thin line between poisonous and healing compounds, dotted across numerous publications, in this review paper.

## 1. Introduction

Members of the genus *Drimia* Jacq ex Willd have been utilized since ancient times for various ailments such as dropsy, respiratory ailments, bone and joint complications, skin disorders, epilepsy and cancer. *Drimia*, also known as *Urginea* (Hyacinthaceae = Asparagaceae) plant, is a distinctive, highly poisonous, deep-red-color bulb native to the surrounding savanna areas of South Africa [1,2], Africa, Asia and Europe [3]. It is a pear-shaped, onion-like, scaly bulb that may grow up to 30 cm in diameter and is found below the surface of the soil [2,3]. *Drimia* species are well known in traditional herbal medicine and used for the treatment of venereal diseases, such as stomach pain, abdominal pains, backache and hypertension, and as a blood purifier and an abortifacient [2,4,5]. 

*Drimia* species and other traditional medicinal plants play an important role in the search for new bioactive compounds. Watt and Breyer-Brandwijk [6] and Hutchings et al. [7] believe that bulbous plants are used to reduce inflammation and possess some antimicrobial activities. In addition, South African traditional healers and herbalists utilize *Drimia* to treat numerous illnesses such as, but not limited to, diseases or problems related to respiratory dysfunction and problems affecting the bones, joints and skin. According to Watt and Breyer-Brandwijk [6] and van Wyk and Gericke [8], traditional healers use the plant to treat venereal infections, abdominal pain, hypertension and numbness and its water extract to purify blood. The bulb is the most used part of *Drimia,* as it can be applied in emetics, heart tonics, expectorants, diuretics and ointments [9]. It can increase sex drive in males and help females who are infertile and ease dysmenorrhea [9]. 

When heated, the bulb scales can treat gout and rheumatic diseases and, when in the form of powder, be used to fight viruses such as influenza and treat chronic diseases such as asthma and bronchitis [10]. Historically, *Drimia* species bulbs have been utilized for the treatment of respiratory conditions and bone, joint and skin complications. In addition, the people of Asia, Egypt and Europe have used *D. maritima* to cure diseases and ailments since 1500 BC. This *Drimia* species is famously known for its healing properties, and it is widely distributed in the Mediterranean area, where it has been commercialized [1]. Traditionally, it has been utilized to treat various ailments such as bladder-related problems, headache, infertility and cardiac edema [11,12]. The underground parts can be administered orally and externally for ritual washing and cleansing [13]. It also has been used for the treatment of retained placenta and related diseases in cattle [14]. Although this plant is used for most ailments in traditional medicine, traditional practitioners are well informed about the toxic nature of the plant.

A number of studies have been conducted on this plant, showing several antiviral, antibacterial, antioxidant and anti-inflammatory biological activities [1,15,16]. Phytochemical screening of *Drimia* resulted in the isolation of several different bufadienolides such as gamabufotalin-3-*O*-a-Lrhamnopyranoside, 12β-hydroxyscillirosidin and arenobufagin-3-*O*-a-L-rhamnopyranoside [17]. In another study, fresh bulbs of *D. sanguinea* also yielded other secondary metabolites such as salicylic acid, scillaren A, phloroglucinol, phlorin, 3-hydroxy-4-methylbenzoic acid and 5a-4,5-dihydroscillaren A [18] with tremendous beneficial properties. However, there is a thin line between the poisonous and healing compounds produced by *Drimia* species. Therefore, in the current review, we conduct an in-depth investigation into the bioactive compounds, toxicology, biological properties and phytochemistry of *Drimia*, concluding on the utilization of such effective therapeutic compounds in drug development. In the same way, studies have isolated bioactive compounds such as phenolic compounds, cardiac glycosides and phytosterol from various *Drimia* species for their medicinal properties [1].

## 2. Methodology Framework

The current review was carried out by searching, screening and selecting research studies and/or papers from five electronic databases, including ScienceDirect, Scopus, Google Scholar, PubMed and Web of Science. Searches were conducted from September 2020 to May 2021. Keywords (“*Drimia*”, “*Urginea*”, “Poisonous”, “Compounds”, “Toxicity”, “Bulbous plants”) were utilized for article selection and screening of abstracts prior to reviewing full publications. Studies focusing on traditional uses, biological properties, in vitro studies and animal and human studies on the genera *Drimia* and *Urginea* were included. Because *Drimia* is a synonym for *Urginea*, they are used interchangeably throughout this review.

## 3. Botanical Characteristics and Geographical Distribution of *Drimia*

*Drimia* is classified as part of the family Asparagaceae (formerly the family Hyacinthaceae). Hyacinthaceae is grouped into 1000 species, with 35 genera currently recognized. Furthermore, the genus *Drimia* belongs to the subfamily Scilloideae, famously known for their bulbous plants. In addition, *Drimia* consists of nearly 100 species. There is an ongoing debate about the classification of the genera *Drimia.* The current consensus from taxonomic research indicates that *Urginea* is a synonym for *Drimia* species. In the Greek language, “*Drimia*” translates to drimus, meaning “acrid” or “pungent”. *Drimia* spp. are deciduous, apomorphic, short-lived flowering plants growing from perennial bulbs. Their seeds are usually black and winged [19,20].

According to de Wet [21], *Drimia* is considered the most important and best-known species among all “slangkop” species. In addition, it is one of the six most common poisonous plants in southern Africa that are toxic to livestock and humans [22]. *Drimia* can grow in various soil types, including lime, clay and stone soils, as well as alluvial soils near rivers or lakes [1]. It has huge, pear-shaped, onion-like, reddish-brown bulbs that are 30 cm in diameter, usually dormant, just below the ground surface. The bulb is covered by a variety of thin black or dark-purple scales that have a bitter taste when applied to the tip of the tongue [23,24,25]. Older bulbs are broken into much smaller bulbs. 

The single flowering stem is normally 30 cm long, but it can grow up to 60 cm and has several small flowers that appear before the leaves in spring. The flowers appear in early spring (September to November), but flowering can begin as early as mid-July in warmer regions. Indigenous bulbs are noticeably wine red with a woody base plant and flashy roots attached [13]. Various species of the genera *Drimia* have been sketched: *D. maritima* [26], *D. elata* [27] and *D. altissima* [28]. Raimondo et al. [29] illustrated different parts of *Drimia sanguinea* under the Red List of South African Plants (SANBI, Pretoria, South Africa) [29].

*Drimia* spp. are widely distributed throughout Africa [1], Madagascar, the Mediterranean area, southeast Asia and India [20]. Studies have recorded that *Drimia* spp. grows in various South African provinces such as the North West, Mpumalanga, Free State, Gauteng and Northern Cape. It is also found in countries such as Zimbabwe, Namibia and Botswana. The geographical distribution of *D. sanguinea* is depicted by SANBI, Pretoria, South Africa [29]. The geographical locations of *D. intricata* [30] and *D. sigmoidea* are in southern Africa, while other *Drimia* spp. are widely spread in the Mediterranean, Spain, Sardinia, Corsica, and Egypt [2].

They are widely known for their medicinal properties, as indicated in the oldest written record from 1500 BC. Until now, the genera *Drimia* have been well investigated for bioactive compounds and commercialized [1].

## 4. Traditional Uses of *Drimia* spp.

Traditional practitioners use *Drimia* because of its therapeutic values and toxic properties [1]. The part of the plant that is utilized in most scenarios is the bulb, as it can be used in many forms, such as in powders, decoctions, ointments, poultice, plant juice, infusions and lotions [31,32]. According to Bozorgi et al. [1] and Asong et al. [16], traditional herbalist practitioners prescribe different parts of *Drimia* to treat, reduce and ease different types of diseases (retained placenta, numbness, dysmenorrhea, venereal infections, rheumatic, abdominal pains, influenza, cancer, headache, respiratory functions, joint and body pain) and conditions such as asthma, low sex drive, bronchitis, gout and infertility. In addition, van Vuuren et al. [13] stated that *Drimia* spp. can be used to perform ritual cleansing. Recent studies showed that the parts of the plant can either be administered orally or topically, and the prescribed dose of the plant material differs depending on the patient’s physical structure and medical conditions, as there is no standardized measurement of prescription. However, traditional herbalists are aware of its nature of toxicity, as described earlier [9]. The brine shrimp lethality assay states that 1 g of the bulb diluted in 10 mL of boiled water has no cytotoxic effect. Recently, it was proven that oral administration is the most frequently used method, followed by topical administration of the plant parts of *Drimia* [33,34,35]. According to traditional Batswana healers, oral administration is mostly used, as most diseases or conditions start from the inside of the body due to a weakened immune system. Traditional herbalists soak different parts of the plant in water for a few days or boil them to extract the bioactive ingredients to be administered orally. They also use it in sprinkling powder, poultice and lotions to be administered topically [8,31,32,36].

Indian squill, known as *Urginea*, is utilized for pharmaceutical and agricultural purposes in the traditional Indian system (Ayurveda). For centuries, ancient Egyptians introduced *Urginea* plants for the treatment of edema, emesis and cough. To date, drug development has used these plant species in present-day medicinal expectorant and cold preparations. In addition, *Urginea* bulb extract was used as a cardiotonic agent by the ancient Romans. Since the 1500s, *Urginea* (*Drimia*) plants gained popularity, resulting in the commercialization of their products, such as digitalis glycoside squill as a cardioactive agent in the United States, prior to approval by the German commission [37].

## 5. Biological Activities of *Drimia* spp.

Cardiovascular effects are the oldest behavior reported among the biological activities of the genus *Drimia*. Previous in vivo or studies in clinical settings of *Drimia* species concentrated on this property. *D. robusta* colonies, both naturally grown and in vitro, demonstrated strong antibacterial and antifungal activity against several pathogens [38], whereby the leaves displayed more strength than the bulb. 

### 5.1. Antimicrobial Activity

To date, antibiotic therapy is still regarded as an essential treatment of secondary infections [39], despite the rise in microbial resistance, hence the evolution of chronic diseases [40,41]. A number of studies have reported antimicrobial properties such as antibacterial, antifungal and antiviral properties of the genera *Drimia* in both in vitro and in vivo models. In addition, reports of the antimicrobial activities of some famous *Drimia* spp. are well documented; however, the antimicrobial activities of other species are limited. Of the 40 medicinal plants tested against 11 strains of bacteria, *D. indica* was graded as successful against bacterial strains. There was also a record of potent activity of *D. indica* against other bacterial strains, especially *Bacillus megaterium* and *Neisseria gonorrhoeae*. *D. sanguinea* bulbs displayed significant anti-*Staphylococcus aureus* activity. It was found that *D. altissima* had no activity against *Listeria monocytogenes*. No significant antiviral activity from the extract of *D. maritima* was reported. Among the active compounds with antimicrobial properties are the homoisoflavanone compound from *D. delagoensis* and scillarenin from *D. maritima* [42]. Baskaran et al. [38] reported that *D. robusta* showed significant antibacterial activity. Furthermore, the utmost concentration (19.68 μg mg^−1^ DW) of proscillaridin A was reported in the roots of ex vitro plants [38]. 

A study conducted on *D. indica* displayed antibacterial and antifungal effects. The minimum inhibitory concentrations (MICs) ranged from 8.2 to 10.6 mg for antibacterial effects and 1.36 to 1.38 mg for antifungal effects. Various bioactive compounds such as salicylic acid, quercetin, coumarins, kaempferol, luteolin and apigenin were isolated from *D. maritima* [43]. Pandey and Gupta [44] extracted the metabolites of *Urginea indica* (*D. indica*) from the roots, stems and leaves using polar (aqueous, methanol), dipolar (acetone) and nonpolar (chloroform) solvents. The plant extracts were tested for antimicrobial activity against *Bacillus cereus*, *Staphylococcus aureus*, *Staphylococcus epidermidis*, *Escherichia coli*, *Proteus vulgaris* and *Pseudomonas aeruginosa* and against two fungi, *Aspergillus niger* and *Candida albicans*. They reported that root methanol extracts exhibited the highest activity against *B. cereus*, while acetone extracts inhibited *P. aeruginosa*. Fungi *A. niger* and *C. albicans* were inhibited by root acetone extract. Furthermore, the phytochemical analysis showed major compounds such as alkaloids, tannins, quinones, saponins, flavonoids, glycosides, phytosterols and resins [44].

In another study, the activity of *U. maritima* bulb extract was used to control foodborne pathogens, including *Listeria monocytogenes*, *Escherichia coli*, *Bacillus subtilis*, *Bacillus cereus*, *Klebsiella pneumoniae* and *Staphylococcus aureus*. The bulb extract was also tested against pathogenic *Colletotrichum graminicola*, *Sclerotium rolfsii*, *Fusarium oxysporum* and *Penicillium digitatum*. The results showed that *U. maritima* bulb extract had the highest antifungal effect *P. digitatum* (EC_50_ = 69.01 ± 2.29 µg/well) and *C. graminicola* (EC_50_ = 86.89 ± 1.17 μg/well). The highest antibacterial activity was detected against *S. aureus* (66.81 ± 1.06%) and *B. subtilis* (57.94 ± 0.92%) [45]. An in vitro propagation study and antibacterial assessments were conducted using *E. autumnalis* and *D. robusta* plants. Several bacterial pathogens, including *B. subtilis*, *Micrococcus luteus*, *Enterococcus faecalis*, *Klebsiella pneumoniae*, *S. aureus*, *P. aeruginosa* and *E. coli*, were used to determine the antibacterial effectiveness of the plant extracts. *D. robusta* bulb extract showed excellent antibacterial properties against *E. faecalis*, *S. aureus* and *M. luteus*. Moreover, *D. robusta* was determined as a noble effective bioresource [46]. Crude extracts of *U. indica* were proven to have good antifungal properties, with high inhibitory effects of 14.06 ± 0.06 mm and 13.26 ± 0.26 mm against *C. albicans* and *A. niger*, respectively [47].

It was discovered that chitinase, which is a hydrolytic enzyme that disintegrates glycosidic bonds in chitin, possessed antifungal properties in the *Urginea indica* (Indian squill) bulbs. The protein was purified, and in vitro results showed inhibitory effects against pathogenic *Fusarium oxysporum* and *Rhizoctonia solani* [48]. Matotoka and coworkers [49] conducted a study to determine the effectiveness of herbal concoctions against HIV reverse-transcriptase and cyclooxygenase activities. *D. elata* Jacq. (Sekanama) was one of the plants investigated. The results proved that concoctions made in combination with *D. elata* extracts exhibited the highest HIV reverse-transcriptase effects (IC_50_ = 2.90 μg/mL), better than current anti-HIV drugs (lamivudine, zidovudine, lopinavir and ritonavir) [49]. In a previous study, Semenya et al. [50] conducted an ethnobotanical survey on indigenous knowledge about plants used to cure sexually transmitted infections by Bapedi traditional healers. It was determined that the *D. elata* bulb is used for the treatment of gonorrhea and HIV/AIDS [50]. Several studies have reported various biocompounds such as scillarenin, tannins, cardiac glycosides and bufadienolides isolated from the genera *Drimia* (*Urginea*) exhibiting good antiviral activities [50,51,52,53,54,55].

### 5.2. Anti-Inflammatory Activity

Aqueous extract of *D. sanguinea* has shown a good range of toxic activity against fungi associated with the deterioration of food commodities and herbal drugs with antiaflatoxigenic activity [56]. The bulb of *D. sanguinea* also possesses an effective antioxidant property [57]. Moreover, a considerable number of antimicrobial compounds have been isolated from extracts of this plant. Several plants have been identified to have anti-inflammatory properties, and most were found to be safe, effective, nontoxic and less toxic anti-inflammatory [58] and antioxidant [59,60]. Current research studies focusing on medicinal plants have been created on the backbone of indigenous knowledge. For example, traditionally, *Urginea maritima* (*D. maritima*) has been utilized for the treatment of cardiac disorders and fungal infections and as a diuretic agent. Hence, Kazemi Rad and colleagues [61] investigated the relaxation effect of *U. maritima* on rat tracheal smooth muscles. It was suggested that the incubated tissues compressing *U. maritima* extract showed significantly higher relaxant outcomes compared to the nonincubated tissues. The bronchodilatory effect of the plant extract promotes the beta-2 adrenoceptor and prevents the muscarinic receptor, potassium opening and calcium channels [61]. The anti-inflammatory activity of *D. nagarjunae* extracts from the leaves and bulbs of the plant using in vitro protein denaturation techniques was investigated further. Nonpolar to polar compounds were extracted by using various solvents, including hexane, chloroform, ethyl acetate, methanol and water. The plant extracts exhibited strong anti-inflammatory activity of 82.97% ± 1.16 at 100 µg/mL [62]. The cardiac glycoside compound showed a potent acute and chronic inflammatory effect and reduced inflammatory symptoms in vivo and animal models [63]. In an assessment of the anti-inflammatory activity of *U. indica* extracts isolated from aqueous and ethanolic solvents, it was shown that these extracts aid in relieving joint inflammation using various models [64].

### 5.3. Antioxidant Activity

Recently, a study evaluated the effects of *D. maritima* flowers and bulbs (essential oils) using various techniques, including DPPH, ABTS+ and total antioxidant capacity. It found that their essential oils possessed excellent antioxidant activity better than Trolox and vitamin E. The nitric oxide chelation scavenging activity of ethanol extracts from bulbs and flowers exhibited IC_50_ values of 5.05 and 5.12 μg/mL, respectively. Analytical GC-MS results showed high levels of eugenol and carvacrol, which account for 41.23% and 27.29%, respectively. *D. maritima* essential oils also demonstrated strong antimicrobial activities [64]. In a 2010 study, Mammadov et al. [65] evaluated the antioxidant activities of *U. maritima* extracts produced from leaves and tubers using different solvents such as methanol, benzene, ethanol and acetone. A β-karotene-linoleic acid system and DPPH assay (free radical scavenging) were utilized to determine the total antioxidant activity of the *U. maritima* extracts, and according to the results, ethanol extracts showed high antioxidant activity of 72.67%, while the benzine extract activity was 31.12%. In addition, methanol extract demonstrated free radical scavenging activity of 66.89% [65]. Mahato et al. [66] found methanolic extract of *U. indica* bulbs possessed remarkably antioxidant activity using DPPH assay with an IC_50_ value of 51.87 µg/mL, which was greater than gallic acid with an IC_50_ value of 39.91 µg/mL. In addition, phytochemical screening revealed several biocompounds with alkaloids as the highest quantity, followed by flavonoids, phenols and saponins. The authors recommended *U. indica* (traditional wild onion) as an alternative for the management of numerous chronic diseases [66].

In a study on the evaluation of the antioxidant and free radical scavenging activity of various *Drimia* sp. (*D. govindappae, D. coromandeliana, D. indica, D. polyantha, D. nagarjunae, D. razii* and *D. raogibikei*) bulb extracts, results demonstrated that *D. coromandeliana* displayed the most antioxidant and free radical scavenging activities [67]. In a recent study, free radical scavenging techniques such as DPPH, superoxide anion, hydroxyl radical and ABTS were utilized to determine the antioxidant activities of *D. maritima*. The ethyl acetate extract demonstrated good scavenging activity and reduced power by employing DPPH and ABTS tests. Furthermore, aqueous extract displayed the highest activity against superoxide anions, hydroxyl radicals and lipid peroxidation [68].

### 5.4. Anticancer Activity

A previous study conducted on the bulbs of South African *D. altissima* isolated novel bufadienolides and drimianins A–G (1–7). Furthermore, a screening assay utilizing bufadienolide showed anticancer activity against human cancer cell lines in the NCI-60 screen [69,70]. On the other hand, novel flavonoid C-apioglucoside, 6-C-[-apio-α-D-furanosyl-(1→6)-β-glucopyranosyl]-4′, 5, 7-trihydroxyflavone (altissimin) was recently discovered from the chemical characterization of *D. altissima.* An in vitro bioassay showed antiproliferative potency against HeLa cervical cancer cells [70]. In a study conducted by Bevara et al. [71], the effect of C-glycosyl flavone was tested on human normal epithelial, breast, hepatic and colon cancer cell lines. The results indicate cytotoxicity potency of C-glycosyl flavone with respect to the induction of apoptosis, cell cycle arrest and inhibition of angiogenesis via CDK6 [71].

Proscillaridin A and cardiac glycosides are among the compounds that were isolated from *Drimia* spp., exhibiting cytotoxic potency and/or antiproliferative activity against human breast carcinoma [72]. In a follow-up study, *D. robusta* extracts were prepared from whole plants and exhibited anticancer effects against human cell lines such as breast MCF7, melanoma UACC62 and renal TK10 [73]. A recent study on *U. maritima* bulb extract also showed anticancer properties by preventing cell cycle arrest and inducing apoptosis in breast cancer cell lines [74]. In an animal study, *U. indica* methanolic extract revealed anticancer inhibitory potential in Swiss albino mice [75]. A number of researchers have determined that *Drimia* species have been utilized for a broad spectrum of applications, including ailments, respiratory conditions, bone and skin disorders, joint complications, cancer and epilepsy. An in-depth in vivo and in vitro investigation showed beneficial antibacterial, antiviral, antifungal, anti-inflammatory, antioxidant and insecticidal effects [1], as shown in Table 1.

## 6. Toxicity of *Drimia* spp.

### 6.1. Toxicity of Drimia spp. in Animals

In animals, intoxication may lead to muscular weakness, paralysis and even fatality caused by heart failure [96]. A number of studies have reported cardiac glycoside as a major toxic compound produced by *Drimia* species [19,99,100,101]. According to Kellerman et al. [23], livestock ingest the flowers, leaves and sometimes the bulb of this plant as substitutes to green grass, and it was found to be toxic to livestock due to cardiac glycoside extracted from the plant’s bulb. *Drimia* species are considered highly poisonous, especially the bulbs of the plants [102]. Due to the toxic nature, several symptoms following administration have been reported such as diarrhea, indigestion, tremors and cramps. However, toxicity caused by this plant in human beings has not yet been recorded; nonetheless, Louw [103] found that some water-soluble chemicals isolated from this plant’s bulbs show toxicity to mankind. The toxicological properties of *D. maritima* were initially described by Theophrastus. Due to its toxic nature, the plant was placed over doors to clear out venomous insects and animals. Furthermore, in traditional Iranian medicine, *D. maritima* plant was also utilized as a mice and insect repellent [15]. This plant has been used to poison mice and prevent mice and insects across the world, specifically in Italy. Its popularity spread across the globe from 1920, and *D. maritima* plant became an effective raticide. Spanish fishermen used the plant bulb as an ichthyotoxic agent. As the word spread, other species within the genera *Drimia* were investigated for their toxic nature by various nations. South Africa recognized *D. sanguinea* plant as one of the six poisonous plants, and in Tanzania, the plant was known as a very toxic plant [15]. Various tribes across the world utilized the plant to detoxify herbs prior to medicinal care. Detoxification by Indian tribes using *D. indica* included saturating the plant in rainwater, boiling it and cooking it. Herbal concoctions of *D. sanguinea* bulbs (ratio of 1:10 (*w*/*v*) of water) prepared by traditional African healers did not have any toxicity on shrimps.

Iranian traditional healers usually add *D. maritima* bulb to *Vicia ervilia* paste and bake it. *D. maritima* bulb is also mixed with honey or water then boiled for the production of squill vinegar and oxymel [15]. *D. sanguinea* bulbs have exhibited significant cytotoxic effects. In a study performed on the mouse fibroblast L929 cell line, aqueous plant extract showed a reduction in cell viability at a 0–2.4 mg/mL dose. However, the number of cells was unchanged. At a 0–1.2 mg/mL dose, *D. sanguinea* extract was potent against cell cultures of chicken embryonic neurons. This activity resulted in the morphological alteration of chick embryo neurons and necroapoptosis, which was familiarized as the primary mechanism. Researchers rationalize the biological potency as a result of biocompounds such as cardiac glycosides, particularly transvaalin. Certain compounds may prevent the Na +/K +- ATPase pump, subsequently leading to an increase in intracellular calcium. Excess of calcium through the mitochondrial permeability transition pathway may result in necroapoptosis. The salicylate compound produced from *D. sanguinea* might stimulate the process. Hence, damage of the axon, neuroglia and astrocytes may occur as a result of an abundance of intracellular calcium [2].

Various cardiac glycosides have been reported as the main contributors to the toxic nature of *Drimia* sp. Furthermore, scilliroside, belonging to bufadienolide glycoside, has been shown to be a toxic compound of *D. maritima* red plants. In an in vivo study, the lethal dose (LD_50_) of scilliroside was 0.7 and 0.43 mg/kg, respectively, for male and female rats. An earlier study showed an LD_50_ for male rats of 0.8 mg/kg and 0.52 mg/kg for female rats [104]. Approximately, at a 100 mg/kg dose, contraction of muscles was recorded after 1–2 h and death after 13.2 h. Scilliroside and its aglycon, scillirosidin, have shown increased toxic levels compared to other bufadienolides such as proscillaridin and desacetylscillirosidin. This may be due to the acetoxy group at the C-6 position of scilliroside. The outcomes of various studies have propelled the use of scilliroside as a rodenticide. The results have shown that female rats are more sensitive to scilliroside than their male counterparts; hence, toxicity is reduced due to the testosterone in the male rats. The outcomes also suggest that mice are killed by scilliroside with smaller quantities compared to rats. The lethal dose (LD_50_) of proscillaridin (a type of bufadienolide) isolated from *Drimia* species was 0.25 mg/kg through oral administration in newborn rats. Moreover, in mature rats, the oral LD_50_ was 56.2 and 76.5 mg/kg for male and female rats, respectively, while intravenous (IV) showed an LD_50_ of 8.7 and 17 mg/kg for male and female rats, respectively. The significant difference between oral and IV toxicity in infants and mature rats means higher sensitivity of young rats to proscillaridin, which has nothing to do with intestinal absorption varieties. Transvaalin was isolated from *D. sanguinea* plants, which was regarded as a principal toxic bufadienolide compound; however, this was identical to scillaren A. The lethal doses of transvaalin (scillaren A) and proscillaridin through the IV route in cats were reported as 0.156, 0.167 and 0.184 mg/kg, respectively. Scillaren is a derivative cardiac glycoside of *Drimia* sp., which results in cholestasis in rat liver by impacting the blood flow [104].

Teshome and colleagues [105] studied *D. altissima* plants to assess the toxicity and palatability of the bulbs by utilizing the rat model obtained from the field. Furthermore, the study aimed to use *D. altissima* plants as potential rodenticide candidates. Various groups (treatment and control) of rats were administered different concentrations of *D. altissima* concoctions in a controlled laboratory environment, and the fatality count was documented during the experiment period. The results showed a large number (80−100%) of fatalities after administration of the poison bait of *D. altissima*. In addition to this, the powdered bulb was responsible for 8%, producing 50% of the fatalities. The researchers concluded that the powdered bulb of *D. altissima* caused fatalities in rat mortality; however, no fatalities were recorded in the control group. Moreover, an in-depth investigation is required to support and expand on the toxicological properties of *D. altissima* bulb and its active compounds [105]. A study conducted on the toxicity properties of *D. maritima* prepared in ethanolic extract was used to investigate the fatality rate, sexual conduct and oviposition behavior of vinegar fly, *Drosophila melanogaster*. The vinegar fly was used as a test model in the laboratory environment. Administration of the treatment was conducted by ingestion of second-stage larvae (L2). Laboratory data indicated that 100% of fatalities were achieved after 15 days of treatment. In addition, there was a significant reduction in the quantity of eggs laid and larvae quantity of the first-generation-treated and partial nuptial courtship. The results showed the neurotoxic effects of *D. maritima* ethanolic extract, suggesting the plants produced toxic secondary metabolites [106].

### 6.2. Toxicity in Humans

Tuncok and coworkers [107] reported that *Drimia* species caused the death of a 55-year-old woman who ingested two bulbs orally due to arthritic pain. Before ingesting this herb, she was diagnosed with Hashimoto’s thyroiditis and hypothyroidism, which might have been associated with serious toxicity. Digitalis-like signs of poisoning included vomiting, seizure, severe hyperkalemia, ventricular arrhythmias and atrioventricular block [107]. Squill opiate linctus, which was applied as a cough remedy in two patients, caused toxic effects. Bradycardia and other signs of digitalis intoxication were found in both cases, and toxicity occurred after an overdose of this drug [108]. The major symptoms of 14 patients with *D. sanguinea* poisoning are gastrointestinal, central nervous system and urinary tract problems. Accidental inhalation of a scilliroside-containing substance under the trade name “Silmurine” resulted in certain effects that were not prolonged, such as headache and vomiting. Bulbs and leaves of *Drimia* species can cause itching and skin inflammation because of their calcium oxalate content [9]. Polat et al. [109] utilized *D. maritima* bulbs as a topical application for arthralgia, which resulted in nonallergic irritant contact dermatitis in a 52-year-old woman. To determine the toxicology of poisonous plants including *Drimia* species, more in-depth studies are required using in vitro and in vivo models in both animal and human clinical investigations. Table 2 indicates some toxicity effects *Drimia* species reported so far in animal models.

From the Google Scholar search tool, a total of 462 publications were retrieved by using keywords “*Urginea*” and “Toxicity” from 2017 to 2021 (5 years). Out of these publications, only three were applicable. Furthermore, screening other electronic databases such as ScienceDirect, Scopus, PubMed and Web of Science showed a lower number of publications. The selection process used a combination of keywords such as “*Drimia*”, “*Urginea*”, “Compounds”, “Toxicity” and “Safety”, and the findings are summarized in Table 3.

## 7. Phytochemicals of *Drimia* spp.

Due to the therapeutic effects of *D. sanguinea* bulb, researchers have focused on the phytochemicals of *Drimia* species, and the leaves and roots have also been examined [1]. The principal constituents isolated from this genus are cardiac glycosides. In addition, in these plants, phenolic compounds, phytosterols and other phytochemical constituents have been identified [112,113]. The major phytochemicals that are commonly present in *Drimia* spp. are alkaloids, tannins, quinones, saponins, flavonoids, glycosides, phytosterols, resins, salicylic acid, quercetin, coumarins, kaempferol, luteolin and apigenin [43,44]. In another study, GC-MS analysis was utilized to identify compounds in *U. indica* crude extract. The results showed 36 compounds were identified, namely 9,12,15-octadecatrienoic acid, stigmasterol, squalene, hypocholesterolemic *n*-hexadecanoic acid, diuretic phytol, pyrogallol 10.40%, 9,12-octadecadienoic acid and octadecanoic acid. In addition, a number of alkaloids, flavonoid glycosides, saponins, proteins and carbohydrates were recognized [52].

### 7.1. Cardiac Glycosides

The positive impact on the function of the heart and blood vessels (cardioactive effect) caused by *Drimia* species has sparked interest in the identification of compounds since the early 1800s [1]. In 1933, Arthur Stoll was the first person to extract scillaren A (cardiac glycoside) from *D. maritima,* which was a novel discovery in cardiac therapy [114]. Cardiac glycosides are organic steroidal compounds consisting of C-24 or C-23 and biological properties such as inotropic and chronotropic effects [115]. Furthermore, the cardiac glucoside structure is composed of tetracyclic 10, a 13-dimethyl-cyclopentanoperhydrophenanthrene nucleus and its steroidal nucleus, which is known by the cell receptors [116,117].

A comparative study was conducted by El-Seedi et al. [92], whereby 61 Egyptian medicinal plants from 29 families were investigated. The study suggested that cardiac glycoside from *U. maritima* was accountable for the cytotoxic activities [92].

Cardenolides and bufadienolides (Figure 1) are chemical compounds belonging to cardiac glycosides, depending on the lactone ring comprising five or six carbon atoms [118,119]. The sugar moiety of cardenolides and bufadienolides affects pharmacological actions. In this regard, the sugar chain consists of one to three sugars, which are connected to Position 3 of the steroidal core [120]. Morsy [121] reported that flowering plants (angiosperms) are an abundant source of cardiac glycosides. Cardenolides are widely abundant compared to bufadienolides, and there is little possibility to obtain bufadienolide from animals and plants. Proscillaridin A (endogenous bufadienolides) has been obtained from mammalian plasma and other body fluids. However, *Drimia* plants are rich in bufadienolide.

Follow-up studies were conducted based on the initial isolation of scillaren from *D. maritima* plant in 1933 [114], and researchers focused on bufadienolide from *Drimia* plants. Nuclear magnetic resonance spectroscopy (NMRS) for the identification of bufadienolides and high-performance liquid chromatography (HPLC coupled with a detector) were also utilized to identify the compounds in a mixture [121].

Analytical techniques were also used for the detection of major constituents, including proscillaridin A, scillaren A and scillirosiden from different varieties of *D. maritima*. In addition to bufadienolide compounds, several cardenolides were isolated from methanolic extract of *D. fugax*, and the structure of new cardenolide was elucidated using high-field NMR spectroscopy. In a recent study, phytochemical investigations showed cardiac glycosides of Indian *Drimia* exhibited significant antioxidant properties, which serves as an ideal candidate for the isolation of bufadienolides. Bufadienolides, namely scillaren A, were isolated from *D. coromandeliana* and *D. razii* [25]. Proscillaridin A (cardiac glycoside) derived from *U. maritima* exhibited antimicrobial [38] and anticancer properties [73]. Several studies have utilized this compound for the treatment of congestive heart failure and cardiac arrhythmia [38,73,122].

An earlier study used fast atom bombardment mass spectrometry (FAB-MS) and nuclear magnetic resonance (NMR) to detect 10 unknown biocompounds (Compounds 6, 14, 17, 19, 22–26 and 32) from the bulbs of *U. maritima*. Compounds 6, 14 and 17 belonged to bufadienolides, which lacked sugars; Compound 32 was lignan glycoside, which is unusual in *U. maritima*; and Compounds 19 and 22—26 were unfamiliar bufadienolide glycosides [123]. Cardiac glycosides are characterized as sugar residue with an unsaturated lactone ring (five or six atoms) and a steroidal residue. These secondary compounds are produced in plants, insects and animals. In the past, plant or animal extracts comprising cardiac glycosides were used for diuretics, emetics, as poison on arrows and darts and for suicide or murder [124].

### 7.2. Phenolic Compounds

A number of phytochemical compounds have been isolated and identified from *Drimia* species, including flavonoids, which are phenolic metabolites [23]. The TLC technique was used to detect the flavonoid compounds from cardiotonic glycosides [125]. Freeform pelargonidin-3-monoglucoside and cyanidin-3-monoglucoside and ρ-cumaric acid acylated with caffeic acid were obtained from Spanish *D. maritima* (red bulbs). Other phenolic compounds were quercetin-3-monoglucoside, taxifolin 4′-glucoside and *C*-glycosyl flavones.

Figure 2 illustrates some chemical structures of phenolic compounds such as taxifolin (C_15_H_12_O_7_), quercetin-3-monoglucoside (C_21_H_19_O_12_) and flavonoid C-glycosyl (C_22_H_22_O_11_) isolated from *Drimia* spp. A homoisoflavonoid compound was isolated from *D. delagoensis*. Three flavonoid glycosides were identified in *D. indica* bulbs. Caffeic acid from *D. maritima*, 4-hydroxy-3-methoxybenzoic acid from *D. delagoensis* and phloroglucinol derivatives from *D. sanguinea* were also identified as other phenolic constituents [1]. Langat et al. [70] discovered a new flavonoid called C-apioglucoside, 6-*C*-[-apio-α-D-furanosyl-(1→6)-β-glucopyranosyl]-4′, 5, 7-trihydroxyflavone, which was isolated from *D. altissima* plants. Moreover, the compound exhibited good antiproliferative potential [70]. Various phytochemicals were isolated and identified from the bulbs of *U. maritima*, and the main findings revealed high concentrations of polyphenols and flavonoids. Furthermore, HPLC-ESI/TOF-MS analysis detected ferulic acid, vanillic acid and 4-hydroxybenzoic acid as the main phenolic compounds. The biocompounds exhibited some beneficial insecticidal properties and restrictive activity on the acetylcholinesterase enzyme system in the rice weevil *Sitophilus oryzae* (L.) (Coleoptera: Curculionidae) [45].

In a follow-up study, various biocompounds such as tannins, phenols and flavonoids were extracted from *U. indica* bulb extracts and identified [66]. A number of secondary compounds were produced from the fresh plant material of red sea squill (*U. maritima*). The plant extract was prepared with aqueous acetone (90:10, *v*/*v*). Reverse-phase HPLC (RP-HPLC) coupled with DAD and MSn detection was used to identify several compounds, including cardiac glycosides, phenolic acids and flavonoids. Dihydroquercetin, which is a potent flavonoid, was detected in high concentrations [126]. In the endless pursuit of novel biocompounds, Sultana et al. [127] discovered three unknown flavonoid glycosides, namely 5,4′-dihydroxy-3-*O*-α-L-rhamnopyranosyl-6-*C*-glucopyranosyl-7-*O*-(6′′-para-coumaroyl-β-D-glucopyranosyl) flavone (2)5,6-dimethyoxy-3′,4′′-dioxymethylene-7-*O*-(6′′-β-D-glucopyranosyl-β-D-glucopyranosyl) flavanone (1), and 5,4′-dihydroxy-3-*O*-(2′′′′′-β-glucopyranosyl-α-L-rhamnopyranosyl)-6-*C*-glucopyranosyl-7-*O*-(6′′-para-coumaroyl-β-D-glucopyranosyl) flavone (3) from *U. indica* bulbs (Indian squill) [127].

### 7.3. Phytosterols

A study showed that beta- and gamma-sitosterol were obtained from *D. indica* bulbs. Furthermore, the leaves, bulbs and roots of different *D. indica* cytotypes were studied, and phytosterols was the dominating sterol, followed by stigmasterol. Campesterol was only obtained from triploides. Stigmasterol was also isolated from *D. sanguinea* bulbs [1]. The chemical structures of phytosterols, including cholesterol, campesterol and stigmasterol isolated from *Drimia* spp., are displayed in Figure 3. Phytosterols, β-sitosterol and stigmasterol were detected in plant parts of *U. indica.* The highest total sterol content was noticeable in the leaf with 23.46 mg/gdw, and the lowest was observed in the bulb with 18.18 mg/gdw [128]. However, Raj and Kameshwari [55] utilized liquid chromatography–mass spectroscopy (LC-MS) and nuclear magnetic resonance (NMR) to determine the biocompounds in *U. wightii* extract. Several secondary compounds, including hexadecanoic acid methyl ester, 1,3,7,11,15-tetramethyl-2-hexadecenol and stigmasterol, were identified and established to possess antioxidant effects [55]. Phytosterols are generally referred to as plant sterols that are similar to cholesterol in structure with distant sidechain configurations. They are triterpenes with a four-ring steroid nucleus, the 3β-hydroxyl group and, frequently, a 5,6-double bond. In addition, the purpose of phytosterols is to balance the phospholipid bilayers in cell membranes. They have various applications in cosmetics, nutrition and therapeutic purposes. Other critical properties include anticancer properties [129].

### 7.4. Miscellaneous Compounds

Two alkaloids were isolated from *D. altissima* [130] and were potent against *Phytophthora capsici*, a tomato pest. The results were possible but not accurate because of the similarities between *D. altissima* and some amaryllidaceous, as there are still a lack of herbarium data and species. From *D. altissima*, *Eudesmane sesquiterpenoids* have been identified. Calcium oxalate needles have been released from *D. altissima* flowering bulbs [131]. Idioblasts from *Drimia* have been reported as causative factors of surface irritation [132].

Histamine was not found in *Drimia* plant. However, calcium oxalate raphides were found in *D. maritima*, particularly mucilaginous idioblasts. Other compounds such as sinstrin were obtained from *D. maritima* plant, which was utilized for renal clearance. Dihydro-benzofuran-typeneolignan glucoside and free amino acids such as L-azatidine-2-carboxilic acid were also isolated from *D. maritima,* together with trace amounts of volatile oils. Approximately 29 kDa glycoprotein, which possessed some antifungal and antitumor effectiveness, was found by gel filtration and reverse-phase HPLC from *D. indica* bulbs. Lectin-like protein and asteroidal sapogenin are other miscellaneous compounds isolated from the bulbs of *D. robusta* and *D. sanguinea*, respectively [1]. Medicinal plants produces a variety of bioactive compounds with different concentrations, thus poking important issues in regards to quality, safety and efficacy [133].

## 8. Conclusions

Human populations have depended on the environment since the beginning of time and utilized plants for fuel, clothing, shelter and food, as well as medicine, for their primary survival. This is evident today, as approximately 80% of the global population relies on plants for their primary health care needs. Historically, knowledge of medicinal plants was verbally transferred from one generation to the next, and limited information was documented. Today, researchers utilize indigenous knowledge to validate the biological properties and toxicity of plants. The current review presented comprehensive information on *Drimia* species, particularly about the biocompounds producing biological activities and toxicity effects. *Drimia* is known as a dark-red bulb plant with poisonous characteristics. However, traditional practitioners use its extracts to purify blood and treat sexually transmitted diseases. Various researchers have established that the genus *Drimia* is used to treat dropsy, respiratory disease, bone and joint complications, skin disorders, epilepsy and cancer due to its numerous secondary biocompounds. The current review clearly shows an increase in biological investigations of *Drimia* spp. Nevertheless, research has focused on the biological benefits of *Drimia*; hence, there are limited data on the toxicity of these plants. Because biocompounds from the *Drimia* genera might have poisonous and healing properties, there is a thin line. In conclusion, there is an existing gap between poisonous and healing properties. Due to the number of fatalities in both humans and animals, we encourage more investigations on the toxicity and safety of these plant species.

## Figures and Tables

**Figure 1 pharmaceutics-13-01385-f001:**
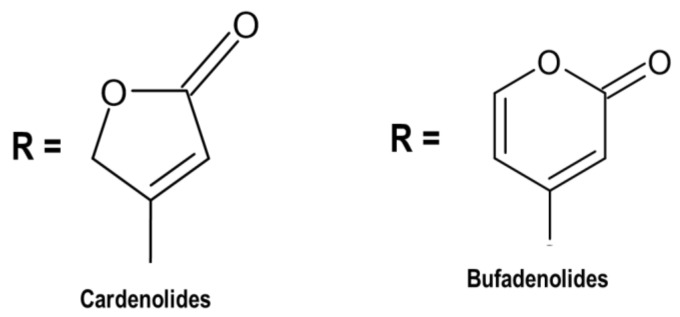
Two classes of cardiac glycosides occurring in nature [118].

**Figure 2 pharmaceutics-13-01385-f002:**
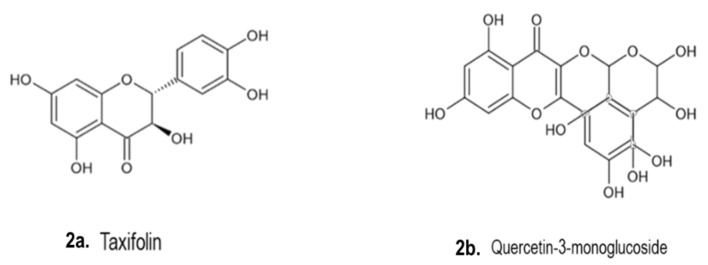
Chemical structures of phenolic compounds isolated from *Drimia* spp. [1]. (**a**). Taxifolin (Type of flavonoid, C_15_H_12_O_7_); (**b**). Quercetin-3-monoglucoside (C_21_H_19_O_12_); (**c**). Flavone (C_15_H_10_O_2_) and (**d**) Vitexin (Type of flavone glycosides, C_21_H_20_O_10_).

**Figure 3 pharmaceutics-13-01385-f003:**
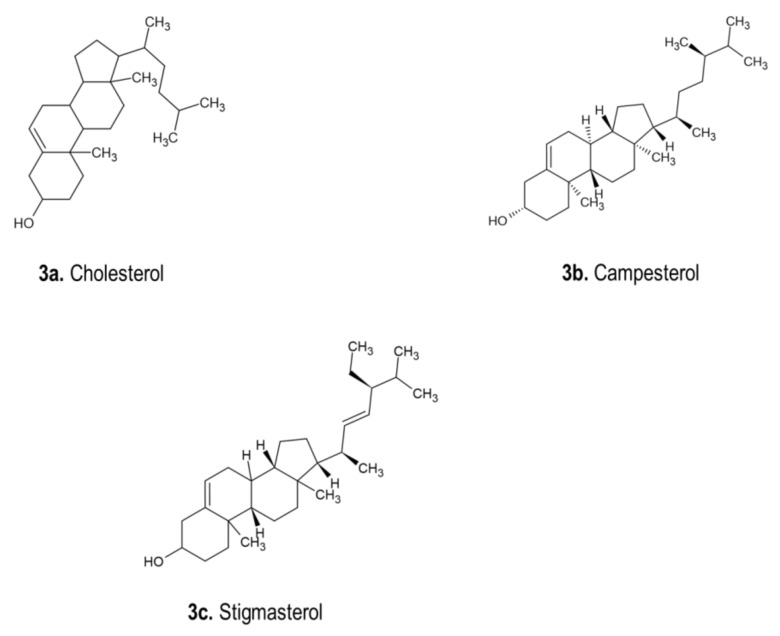
Chemical structures of phytosterols isolated from *Drimia* spp. [1]. (**a**). Cholesterol (C_27_H_46O_); (**b**). Campesterol (C_28_H_48O_); (**c**). Stigmasterol (C_29_H_48O_).

**Table 1 pharmaceutics-13-01385-t001:** Biological investigation of *Drimia* plants.

*Drimia* Species	Biological Activity	Part Used	Extraction Form	Chemical Composition	Ref.
*D. sanguinea*	Antibacterial activity, antifungal, antioxidant, anticytotoxicity	Bulbs	Methanol extract, petroleum ether, extract	Pentanoic acid, *n*-hexadecanoic acid, 1-nonadecene, hexadecanoic acid, ethyl ester, di-isooctyl phthalate, α-sitosterol	[16]
*D. indica*	Antifungal activity	Bulbs	Crude extract, methanol extract	*O*-glycosyl flavanone, *O*-glycosyl flavone and *C*-glycosyl flavone	[43]
*D. indica*	Anthelmintic activity	Bulbs	Aqueous extract, crude extract	Not specified	[43]
*D. indica*	Antitumor activity	Bulbs	Crude extract	a *C*-glycosyl flavone (5,7-dihydroxy-2-[40-hydroxy-30-(methoxymethyl) phenyl]-6-*C*-βglucopyranosyl flavone	[76,77]
*D. indica*	Antibacterial activity	Bulbs	Aqueous extract, ethanol extract, methanol extract	Not specified	[78,79,80]
*D. indica*	Antioxidant activity	Bulbs	Methanol extract, chloroform extract	Flavonoids, phenolic and proanthocyanidins	[81,82,83]
*D. coromandeliana, D. govindappae, D. indica, D. nagarjunae, D. polyantha, D. raogibikei* and *D. razii*	Antioxidant activity	Bulbs	Hydrochloric acid extract and methanol extract	Total phenolics and proanthocyanidins	[68]
*D. indica*	Antidiabetic activity	Bulbs	Ethanol extract	Not specified	[84]
*D. indica*	Anti-inflammatory	Bulbs	Alcoholic extract	Not specified	[85]
*D. robusta*	Antibacterial, anti-inflammatory, antihypertensive and anticancer activities	Leaf Bulb	Alcoholic extract	Cardiac glycosides, bufadienolides	[86]
*D. maritime*	Antibacterial, anti-inflammatory and anticancer activities	Bulbs	Ethanol extract	Cardiac glycosides, bufadienolides sclerosis and triterpenoids.	[87]
*D. robusta*	Antibacterial and anticandidal activities	Bulbs and leaves	Petroleum ether, dichloromethane, ethanol and water extracts	Phenolic compounds	[88]
*D. maritima*	Antimalarial activity and cytotoxicity	Bulb	Aqueous extract	Not specified	[89]
*D. maritima*	Asthma effect	Bulb	Squill oxymel (a traditional form of *Drimia maritima*), simple oxymel	Not specified	[90]
*D. maritima*	Anticancer effects	Whole plant	Methanol extract	Cardiac glycoside	[91]
*D. maritima*	Analgesic effects	Squill bulb	Proscillaridin A, taxifolin and scilliroside	Not specified	[92]
*D. maritima*	Antioxidant activity and antihemolytic effect	Flowers	Ethanolic, chloroform and ethyl acetate extract	Total phenolic, flavonoid and tannin	[68]
*D. robusta*	Antibacterial activity	Bulb	Ethanolic extract	Cardiac glycosides (2-deoxy sugars), bufadienolides	[93]
*D. maritima*	Antioxidant activities	Leaves and tubers.	Ethanol, methanol, acetone extracts	Phenolic compounds	[68]
*D. macrocentra* and *U. riparia*	Anticancer activity	Bulbs	Extracts	Bufadienolides, rubellin and riparianin	[94]
*D. maritima*	Acaricidal activity	Leaves and bulbs	Methanol, ethanol, acetone and butanol	Bufadienolides derivatives	[95]
*D. numidica*	Antioxidant activity	Flowers, scales, leaves, bulbs and roots	Methanolic	Bufadienolides and total phenolic content	[96]
*D. nagarjunae*	Anticancer activity	Bulbs and leaves	Ethyl acetate and chloroform	Acetic acid, (D,L)-malic acid, hexadecanoic acid, ethyl[4-t-Butyl-2,6-bis(1-methoxy-1-methylethyl)phenyl]phosphinate, octadecanoic acid, 2-hydroxy-1-(hydroxymethyl)ethyl ester	[62]
*D. maritima*	Decreasing dyspareunia and increasing sexual satisfaction	Squill oil	N/A	Flavonoids	[97]
*D. maritima*	Acaricidal activity	Leaves and bulbs	Methanol, ethanol, acetone and butanol	Bufadienolides	[98]

**Table 2 pharmaceutics-13-01385-t002:** Toxicological effects of *Drimia* species.

*Drimia* Species	Toxicity Effects	Model Used	Compounds	Ref.
*D. indica*	Antidiabetic effects, acute toxicity	Rat model	Not specified	[84]
*D. robusta*	Hemolytic activity induces toxicity	In vitro	Saponins, proscillaridin A,	[93]
*U. maritima*	Mortality rate was 10 mg/mL after 48 h	Mite	quercetin, kaempferol and bufadienolides	[65]
*D. altissima*	Rodenticides 80−100% fatalities	Rat model	Not specified	[105]
*D. pancration*	Repellent activity and contact toxicity	Stegobium paniceum beetles	Steroidal saponins	[110]
C-glycosyl flavone of *D. indica*	No mortality at 250 mg/kg bw; nonetheless, 50 and 100% mortality was detected at 500 and 1000 mg/kg bw	Mice	*C*-glycosyl flavone	[71]
*D. maritima*	10% crude extracts resulted in 100% mortality in larvae and 48% mortality in adult beetles. LC_50_ and LC_90_ values were 16.6 and 34.4 g/L, respectively	Larvae and beetle	Not specified	[111]

**Table 3 pharmaceutics-13-01385-t003:** Results of the Database Search on *Urginea, Drimia* and toxicity from 2017 to 2021.

Search Tool	Google Scholar	ScienceDirect	Scopus	PubMed	Web of Science
*Urginea*	462 (3)	56 (1)	3 (0)	8 (1)	7 (2)
*Drimia*	308 (2)	49 (1)	6 (3)	6 (2)	9 (3)
Numbers in brackets (applicable publications)					

## Data Availability

Not applicable.

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
