# Peer review of "Bulbous Plants Drimia: “A Thin Line between Poisonous and Healing Compounds” with Biological Activities"

_pharmaceutics, 2021, doi:10.3390/pharmaceutics13091385_

Round 1

Reviewer 1 Report

The review concerns the pharmacological properties of the plants belonging to the genus Drimia (family Asparagacea) traditionally used in folk medicine in South Africa. The article described all the sorts of useful biological activities, poisonous properties and phytochemical data concerning different classes of biological active compounds such as cardiac glycosides, phenols etc. The review seems to be comprehensive because the authors analyzed all possible literature sources from 2017 to 2021 by main data bases such as Scopus, Science Direct et other ones. The review is interesting and may be useful for the readers however have serious flaw.

The review is too poor from chemical point of view for publication. All the substances noted in the review should be numbered by unique numbers using through all the text. All these substances should be presented as chemical formulae at separate figures. The use of any substances numbers from original articles are senseless and should be deleted. The authors noted position numbers for atoms and functional groups but didn’t show them on the corresponding formulae (if they presented of course).

Th phrase at the page 16 (Lines 172–175) sems to be senceless: “Most of the researchers applied nuclear magnetic resonance spectroscopy (NMRS) for the identification of bufadienolides; however, some studies utilized high performance liquid chromatography (HPLC coupled with which detector) to perform the task [119]”. The use of NMR for identification of mixtures of natural products without their separation is very rare and the isolation of individual pure substances is necessary as a rule. The identification by HPLC only is not sufficient and further NMR studies are also necessary. I recommend to clarify this position.

The review should be redrawn along the recommendations above. My opinion: major revision.

Reviewer 2 Report

Having reviewed the paper entitled :Bulbous plants Drimia: "A thin line between poisonous and healing compounds” with biological activities,  the authors should consider the following points: 

The proposal of the review is interesting, but the manuscript is quite general, incomplete and disorganized. Some of the botanical descriptions in section 3 can be found in any botanical book.

Figures 1 and 2 are not original and the authors do not state their origin.

The biological activity of Drimia species is very repetitive and disorderly. Activities are mixed and sometimes also with compounds.

The authors do not describe the cardiotonic activity of some of these species in the biological activity section.

In the section on Phytochemicals of Drimia spp. ,the authors within each section should order the different phytochemical groups. For example, within the phenolic compounds: phenolic acids, flavonoids (flavones, flavonols, flavanones, chalcones...), tannins...

Figure 3 does not correspond to a general structure of cardiac glycosides. For example Drimia maritima has its main cardiotonic glycosides with hexagonal lactone.

In Figure 4 the proposed structure as C-glycosyl flavone is also not a general structure. It is a specific flavone, but the authors do not say what it is called.

Reviewer 3 Report

The review “Bulbous plants Drimia: A thin line between poisonous and healing compounds with biological activities” summarizes the progress of the studies regarding Drimia bioactive compounds, toxicology, biological properties, and phytochemistry thus concluding on the utilization of such effective therapeutic compounds in drug development.

The manuscript is well written and it could represent a reference for more systematic researches on Drimia biochemicals and biological functions applied to different kind of supplements or as component in functional foods. I have no comment.

Author Response

No corrections.

Round 2

Reviewer 1 Report

The authors have replied: The chemical structures were drawn on chemSketch to improve the quality of the pictures. Unique numbers as well as their chemical formulae were assigned to the structures. The review show case a representation of the identified compounds from the Drimia plants. This section was to prove to the reader that various compounds were identified not to emphasis on the structure of the compounds. Countless studies have utilized such approaches in similar reviews.

My own comment: There are no any numbers unique for each cited substance. It is impossible to discuss the biological active substances without their chemical structure. Yes, there is a lot of similar reviews that are a sort of scientific literary garbage but I never let pass similar sort of reviews to publication.

Most of the researchers applied nuclear magnetic resonance spectroscopy (NMRS) for the identification of bufadienolides; however in a mixture of compounds, some studies utilized high performance liquid chromatography (HPLC coupled with which detector) to identify the compounds even though further analysis is required [119].

My own comment: I don’t understand the sense of this phrase.

My general opinion: the authors don’t wish to adopt my recommendations and improve the manuscript. Hence, the manuscript should be rejected.

Reviewer 2 Report

From my point of view, the work is still messy and has quite a few errors.

Figure 3 is not the general structure of cardiac heterosides. I recommend that the authors review a book on Pharmacognosy.

In Figure 4, the general structure of the C-glucosyl flavones is also not correct. All C-glucosylflavones do not have glucose and and hydroxyls joined to the same carbon.

In section 7.2: (Phenolic compounds), alkaloids and saponins, that are not phenolic compounds, are named.

There is no section within the pharmacological activity referring to the cardiotonic activity of the heterosides bufanienolides.

Round 3

Reviewer 1 Report

The subject of the article is interesting but the authors attempt to deceive the referee and editors instead of realizing the recommendations. The current style of the review without adequate presentation of all the structures as chemical formulae with unique reference number for each compound is not satisfactory. The authors seem to be too lazy or incompetent in order to create a normal review article and attempt to avoid the laborious work on adequate organization and presentation of the information. The figures 4 and 5 (with numbering) are very strange because present the structures twice – with numbering and without numbering. This numbering is non-necessary and reflects the attempts to deceive the referee and editors. Nevertheless, if the editors allow the publication of a review concerning biological activities of any natural products without normal presentation of information concerning their structures, the manuscript may be published after minor revise (deleting the duplicated formulae from the figures 4 and 5).
